# Cognitive Patterns and Problematic Use of Video Games in Adolescents: A Cluster Analysis

**DOI:** 10.3390/ijerph20247194

**Published:** 2023-12-18

**Authors:** Ignacio Fernández-Arias, Marta Labrador, Mónica Bernaldo-de-Quirós, Francisco J. Estupiñá, Marina Vallejo-Achón, Iván Sanchez-Iglesias, María González-Álvarez, Francisco J. Labrador

**Affiliations:** 1Faculty of Psychology, Complutense University of Madrid, 28223 Madrid, Spain; marlabra@ucm.es (M.L.); mbquiros@psi.ucm.es (M.B.-d.-Q.); fjepuig@ucm.es (F.J.E.); mvalle02@ucm.es (M.V.-A.); i.sanchez@psi.ucm.es (I.S.-I.); maria_gonzalez@ucm.es (M.G.-Á.); labrador@cop.es (F.J.L.); 2University Clinic of Psychology, Complutense University of Madrid, 28223 Madrid, Spain

**Keywords:** video games, addiction, clusters, cognitive patterns

## Abstract

Background: Video game playing (VGP) is an increasingly common leisure activity among children and adolescents, although in some cases, it is accompanied by problems due to misuse. Method: A sample of 2884 children and adolescents aged between 12 and 20, representative of the Community of Madrid (Spain), were studied using a cluster analysis to explore the existence of cognitive patterns associated with engagement, attitudes, and concurrent cognitions. We also explored the relationship between these patterns and problematic VGP, using the 2173 gamers as a reference. Results: The concurrent cognitions were not qualitatively different between the problematic users and the others. High engagement and high activation of concurrent cognitions (intensity and frequency) showed the greatest relationship with problematic VGP. Conclusions: The results suggest the existence of different groups of gamers and the relevance to include psycho-educational aspects in intervention programs, as well as the training of specific skills, especially those related with the control of activation. Limitations related to the sample size and potential supplementary analyses are acknowledged.

## 1. Introduction

The increase in the playing of video games (VGs) and the participation of children and adolescents (CaA) has caused great social concern, especially with arguments that they spend too much time on VGs, estimating weekly averages of between 9 and 40 h [1,2,3]. Also highlighted is an important association between the time spent on video game playing (VGP) and the presence of problems [4,5,6], especially psychosocial and psychopathological problems; changes in academic performance [1,4,5,7]; and health problems, such as low quality and quantity of sleep, not eating properly or a poor and unhealthy diet, injuries, etc. [8].

The publication of the DSM-5 [9] with the inclusion of Internet Gaming Disorders (IGDs) in Section III is a point of reference in the study of this problem [10] as it establishes nine criteria for diagnosing IGD, at least five of which must be met over a 12-month period [9]. This does not mean to say that there is no debate surrounding the DSM-5 proposal to characterise the problem [11]. In fact, epidemiological studies on IGD (or previous names such as VGP addiction or abuse, etc.) provide very different figures on the CaA affected, which range between 1.2% and 10.3% depending on the definition of the disorder (IGD, addiction, etc.), the measurement instrument used, and the country being studied [1,12,13,14,15].

It seems necessary to go beyond the symptomatic description and characterisation of the CaA that use, abuse, or show pathological behaviours related to VGs. Identifying why they play and the factors associated with the development of an IGD will help facilitate preventive and therapeutic tasks in this regard.

The study of why a psychological disorder develops usually begins with the analysis of sociodemographic variables. In the case of IGD, a clear and repeated result has been found; men play much more than women and have more problems with VGP [1,4,16]. Other variables such as age, education level, and socio-economic level do not seem to indicate any differences. Some more specific aspects also point to differences, such as a poor parent–child relationship, which is associated with a greater severity of problems with VGP, or an adequate parental bond that seems to be a protective factor against problems with VGP [17].

Apart from the sociodemographic variables, it is common to study differences in personality characteristics. Although some studies point to the influence of certain characteristics on the preference for certain games [18,19,20], there is no consensus on a direct relationship between specific characteristics and IGD. In summary, this line seems to lead to considerations that are somewhat diffuse and questionable operationally.

An alternative line has focused on motivational aspects, such as those related to the addictive potential of VGs and the rewards they provide [19,21]. Affiliation and socialisation motivations have also been highlighted, especially in VGs like massively multiplayer role-playing games (MMORPGs). Thus, Forrest, King [22] pointed to functional detachment as a factor positively associated with VGP problems, concluding that people with healthy social relationships are less likely to develop these problems. It is generally emphasised that the social motivations for using VGs are more typical of controlled gamers than of “hard” or uncontrolled gamers [23]. In a more specific analysis of motivational factors for VGP, [16] outlines five: (1) To be the best gamer (achievement); (2) To interact with other players (social); (3) To explore the virtual world (immersion); (4) Harmonious passion (controlled playing that does not interfere with life’s responsibilities); and (5) Obsessive passion (uncontrolled playing that often has negative consequences), the latter being the one most related to IGD.

Problems with VGP have also been repeatedly associated with immersion in the game, escaping reality, or obtaining better content and rewards in the game [21,24,25]. However, factors such as impulsivity or the capacity for emotional regulation have led to inconclusive results. The work of Gentile, Choo [26], a longitudinal study with 3034 school children from Singapore over two years, highlights that the frequency of VGP, impulsivity, and social skills could predict problems with VGP developing two years later.

A special line has focused on cognitive aspects, given their relevance in related problems such as Gambling Disorder (GD) [9]. In fact, many of the diagnostic criteria for IGD are similar to those for GD, and many of them also refer to cognitive aspects [27]. Some studies suggest that problems with VGP continue due to users constantly thinking about playing VGs and excessive confidence in their ability to meet certain needs such as achievement, social belonging, and self-esteem [28,29]. Problematic VGP is also associated with cognitive aspects such as irrational thinking, beliefs, and expectations or cognitive biases towards VGs [30,31]. Thus, Forrest, King [22] point out that preoccupation, thinking in a constant or ruminant way about the game, and/or planning the next participation in a game may be good criteria for diagnosing IGD, although they do not specify which cognitive processes lead to the development of problems with VGP. In this area, King and Delfabbro [29] considered that the relationship between cognitive factors and IGD may be more complex than a simple “preoccupation” with gaming (criterion A of the IGD). Four basic factors stand out, which are related to the DSM-5’s nine diagnostic criteria for an IGD: (1) beliefs related to rewards, (2) inflexible and maladaptive rules about the game, (3) self-esteem based on the game, and (4) playing as a way to obtain social acceptance. Subsequent studies have confirmed the relationship between these factors and problematic VGP, emphasising specific cognitions involved in playing behaviour, which explain 28% of the variance of VGP problems [12,22].

The way that they have been reached is also a feature of these models. In some cases, it is from theoretical conceptualisations or derivations of explanations for future problems; in others, from the analysis of empirical data extracted with correlation models. These models allow us to establish predictive relationships between cognitive elements, taken separately or together, and VGP behaviour. The use of techniques such as cluster analysis are of particular importance. They allow us to detect profiles or patterns from a set of variables, enabling a better understanding of the phenomena and allowing the limitations of linear relationships to be overcome. It is clear that the resulting clusters depend on the variables considered, so various clusters have been used in the literature, for example, “hard” vs. “moderate” gamers as well as “occasional” vs. “peripheral” gamers (they only interact with gamers). Bateman and Boon [32] identified four clusters according to gaming behaviour: (1) the Conqueror, who is focused on winning and “beating the game”; (2) the Manager who plays the game as a strategic and tactical challenge; (3) the Vagabond who seeks enjoyment and a fun experience; and (4) the Participant. Bartle [33] also identified four types: winner, explorer, socialiser, and “killer”. Other authors focus on identifying clusters based on the frequency or form of VGP [34], the level of interference generated by the gaming pattern [14], or the coping strategies of gamers, identifying that 58% of gamers with problems show a maladaptive method of coping with frustration [35].

Given the importance of attitudes and cognitions in understanding VGP behaviour and its problems, one thing that stands out is the limited number of studies focused on identifying cognitive patterns or clusters that allow for the characterisation of gamers and, ultimately, the optimisation of prevention plans and treatments. Under these conditions, it is especially interesting to know whether the CaA showing signs of problems with VGP have attitudes and cognitions qualitatively different to those of the CaA who do not. We also want to establish whether the differences are in fact quantitative and thus dependent on the level of cognitive activation. Given these considerations, the objectives of this study are to identify clusters related to (1) the cognitions and attitudes of CaA towards video games, (2) their engagement with VGs in terms of self-observation, and (3) the self-dialogue established in this regard. We are also interested in exploring the relationship between these patterns and VGP problems and their contribution to explaining the total variance.

## 2. Materials and Methods

Table 1 shows the sociodemographics and gaming behavior of the sample. 2887 pupils were selected from 37 schools in Madrid through random sampling stratified by postal district, according to population density, type of school (public, subsidised, and private), and stage of education. The selected schools included stages of compulsory secondary education (ESO, Educación Secundaria Obligatoria, in Spanish) and higher education (bachillerato, in Spanish). High schools in Spain also encompass intermediate and higher-level vocational training studies. This is the reason why the age range extended to 20 years, although participants over 18 years old constituted a minimal percentage. A total of 42.5% of people in the sample were females. The mean age was 15.35 years old and 75.3% stated that they used VGs. The average number of hours spent each week on video games was 6.18 with an average video game use of 3.41 days per week. The majority of the sample played on video consoles (32.8%) or their mobile phones (30.1%) on action/adventure games (24.7%) and massively multiplayer online games (17.3%). Using as a reference those participants who admitted to playing video games (2173), 3% (*n* = 65) met the criteria for IGD by scoring 5 points on at least 5 questions of the IGD9SF [36,37,38].

The deontological questions in the study were audited by the ethics committee of the Psychology Department at the Complutense University of Madrid. A team of postgraduate psychology evaluators, specifically trained in the protocols of this research, went to the schools to carry out the assessment once the relevant informed authorisations and consents had been obtained. The assessment was carried out using the Gamertest tool on the computers of each school [39]. Gamertest is an online expert system developed by our research group whose objective is to detect the problematic use of VG. It includes an entry page and two assessment protocols. The first consists of 4 sections: (1) Demographic data; (2) Assessment of video gaming habits; (3) Assessment of the risk level of VGP problems using IGDS-9SF [37]; (4) Engagement with the game. The second consists of 5 sections to assess: (1) level of self-control/impulsivity; (2) personal, social, school, and family conduct; (3) attitudes and cognitions towards VGs; (4) motivation to change; and (5) health status through GHQ-12 [40]. The Gamertest can be found at: http://www.famgi14.es/gamertest/index.html (accessed on 1 September 2023)

The assessments were in a group, anonymous, and lasted for around 30–40 min. The responses of the participants were collected and coded directly into the database created in this regard.

### 2.1. Measurements

#### Socio-Demographic Variables: Age and Gender

*Video game usage variables*: average hours and days of the week for which they play VGs, the preferred device to play on, and the preferred game.

*Problematic gaming index:* An independent Spanish translation of the IGDS-9SF [37] was used, an instrument based on the DSM-5 diagnostic criteria for IGD [9]. The IGDS-9SF scores were considered to be a continuum, with lower scores indicating the lesser presence of VGP problems. Cronbach’s alpha of this instrument was 0.84.

*Cognitive variables*: Three measurements groups were considered: (1) engagement with the game, which consists of 5 items (Cronbach’s alpha = 0.76); (2) attitudes towards the game, which consists of 6 items (Cronbach’s alpha = 0.77); and (3) concurrent cognitions while playing, which consists of 16 items (Cronbach’s alpha = 0.92). A Likert-type response format of 0 to 4 was used (Never/Rarely/Sometimes/Often/Always).

### 2.2. Data Analysis

Descriptive statistics were produced to characterise the sample. For the scales of engagement with the game, attitudes towards the game, and cognitions whilst gaming; a K-means non-hierarchical clustering, with the number of clusters established in advance as three, was carried out over the sample set (*N* = 2887), identifying the differences between clusters through ANOVAs. Once the clusters had been identified, only the sample of subjects who declared that they played video games (*n* = 2173) was selected, and the association of each cluster with the problematic gaming index (IGDS-9SF) was analysed.

To verify the weight of these clusters in the explained variance of problematic gaming, a custom Univariate General Linear Model was produced with the problematic gaming index (IGDS-9SF) as the dependent variable and the clusters resulting from the previous analysis as fixed factors. Analyses were carried out on the main effects and the interaction effects, and the effects of age and gender were controlled for, including them as co-variables.

## 3. Results

### 3.1. Engagement

From Figure 1, it can be seen that cluster 3, “problem-aware” (Engagement-3), shows higher scores than the other two clusters in self-observation of their own game behaviour in all items. In contrast, cluster 1, “no engagement” (Engagement-1), shows the lowest scores in all items. Cluster 2, “not problem-aware” (Engagement-2), presented a pattern of VGP with high scores in “frequency of gaming for longer than expected” and “frequency of having lost the sense of time” and low scores in the rest of the items. When analysing the problematic gaming index by cluster, it was observed that Engagement-3 showed a mean IGDS-9SF (22.73; *Sd* = 7.14) that was statistically higher (*F*_(2, 2170)_= 535, 47; *p* < 0.001) than that of Engagement-2 (15.49; *Sd* = 4.68) and Engagement-1 (12.68; *Sd* = 3.95).

### 3.2. Attitudes

In Figure 2 it can be seen that cluster 1, “game rejection” (Attitude-1), shows high scores in negative attitudes towards VGP. In contrast, cluster 2, “game acceptance” (Attitude-2), shows high scores in positive attitudes towards VGP and low scores in negative attitudes. Finally, cluster 3, “game acceptance with problem awareness” (Attitude-3), shows high scores in both positive attitudes towards video games and negative attitudes (“playing video games is like a drug and in the long run, you get hooked” and “playing video games is a problem”). Attitude-2 and Attitude-3 showed higher scores in the IGDS-9SF (16.20; *Sd* = 5.64 and 17.57; *Sd* = 6.91, respectively) than Attitude-1 (13.78; *Sd* =5.38) (*F*_(2, 2170)_ = 54.44; *p* < 0.001).

### 3.3. Cognitions

With regard to cognitions or self-dialogue (see Figure 3), cluster 1, “no internal dialogue” (Cognition-1), shows no/low self-dialogue. Cluster 2, “moderate internal dialogue” (Cognition-2), and cluster 3, “High internal dialogue” (Cognition-3), have a similar self-dialogue pattern, but it is more intense in Cognition-3, especially the item “I will not stop until I beat this level. Better now than tomorrow”. The IGDS-9SF score was significantly higher in Cognition-3 (24.1; *Sd* = 7.57) compared to that in Cognition-2 (16.58; *Sd* = 4.68) and Cognition-1 (12.21; *Sd* = 3.46). These latter two also differed significantly from each other (*F_(_*_2, 2170)_ = 688.08; *p* < 0.001).

Table 2 shows the estimated marginal means of each cluster and the contribution of the IGDS-9SF to the variance, with age and gender controlled for as co-variables. The explained variance was 50.6% (adjusted *R*^2^ = 0.501). All the variables and interactions included in the model were significant (*p* < 0.05), except attitudes and their interaction with engagement (*p* = 0.083 and *p* = 0.272, respectively). The greatest explained variance corresponds to the engagement variable (partial *eta*^2^ = 0.091; *F*_(2, 2172)_ = 108.28; *p* < 0.001). Specifically, Engagement-3, “acceptance with problem awareness”, showed a higher estimated mean (21.54; *Sd* = 0.33) than Engagement-2, “acceptance with no problem awareness” (16.09; *Sd* = 0.20), and Engagement-1, “rejection” (15.34; *Sd* = 0.41). The clusters of cognitions or self-dialogue obtained the second-highest contribution to IGDS-9SF (partial *eta*^2^ = 0.072; *F*_(2, 2172)_ = 83.96; *p* < 0.001). Cognition-3 showed a statistically higher estimated marginal mean (21.89; *Sd* = 0.47) than the other two clusters.

Interactions between cognitions and engagement (partial *eta^2^* = 0.014; *F*(_4, 2172)_ = 7.62; *p* < 0.001) as well as between engagement and attitudes (partial *eta^2^* = 0.005; *F*_(4, 2172)_ = 2.53; *p* = 0.038) were significant, although with a modest impact on the model. No significant interaction was detected between the attitudes and cognitions.

## 4. Discussion

This work aims to assess the existence of cognitive, attitudinal, and engagement differences in VGP and, if they exist, to see if they are related to VGP problems. Three profile clusters were developed for the engagement, attitude, and cognitions variables, considering the 2887 participants. Secondly, the association between each of these clusters and the level of problematic gaming was examined (IGD-9SF), using as a reference the 2173 subjects who declared that they play video games. 

With regard to engagement with the game, the presence of three significantly different clusters was observed, showing the relationship between the level of problematic gaming and the effect to be of a significant size. The Engagement-1 cluster (no engagement), with low scores in all items on the scale, highlights that neither the gamer nor the people important to them identify anomalous behaviours in their VGP. They seem to have little engagement with VGP and, consequently, have no awareness of problems with it. The Engagement-2 cluster (engagement with no problem awareness) shows a self-perception of anomalies in the way they game (gaming for longer than planned or losing track of time when gaming), but this realisation, not reinforced by other people, is not enough to be aware of problems with VGP. The Engagement-3 cluster (engagement with problem awareness) presents slightly higher scores in the items on their perception of behavioural anomalies in their VGP, but the most relevant thing is that they think that they have problems with their VGP both because other people have told them so and because they believe it themselves. It is possible that this greater awareness of the problem is related to the slight increase in the three anomaly items (gaming for longer, losing track of time, and not being able to meet with friends), although the differences are small, or because this is suggested by people important to them. 

The scores referring to levels of problematic gaming (IGD) are ordered following this progression: lower in those who do not observe anomalies in their VGP behaviours (Engagement-1), higher in those who observe them but have no problem awareness (Engagement-2), and highest in those who observe them and have problem awareness (Engagement-3), although the differences are only significant between the Engagement-3 cluster and the other two, but not between these other two, which seems to highlight the importance of problem awareness. It therefore seems that in VGP problems, as occurs in other addictions, the observation of anomalous behaviours is relevant in becoming aware of the problem, but these behaviours have to be of a certain intensity and must be reported by people important to the player. The fact that there are only differences in the IGD scores between the Engagement-3 cluster and the other two may highlight that a very high magnitude of anomalous behaviour is needed for problems to appear, perhaps as a cut-off point rather than a straight-line relationship between gaming problems and their frequency of use [26,41]. It would be interesting to know whether both aspects are necessary, a minimum level of behavioural anomalies in VGP and the considerations of people important to the gamer, for “problem awareness” with VGP, or whether one of them is sufficient on its own. 

The results seem to indicate the importance of facilitating the “perception of anomalies in VGP behaviour” as a preliminary step to the development of “problem awareness”, which may be helped by the considerations of important people, perhaps as “reality checks”. Once they begin to think that what they are doing is “anomalous”, it is easier to progress to problem awareness. Once this problem awareness has been developed, which is a basic starting point in any intervention for problems with addiction, it will be necessary to provide them with specific skills to manage their VGP behaviours. To summarise, the data seem to support proceeding with VGP problems in a similar way to that advised for other addictions, considering the stages-of-change approach proposed by Prochaska and DiClemente [42,43].

The clusters related to attitudes towards video games do not appear to have a significant relationship with the level of problematic gaming. These figures provide little support for studies finding that favourable attitudes towards video games may be a risk factor and negative attitudes may be a protective factor [12]. We can see that the Attitude-1 cluster, which highlights the negative aspects of VGP, achieves scores in the IGD-9 similar to those of the other two clusters. A diachronic analysis, allowing us to see whether this situation of game rejection (Attitude-1) is reached before starting to play or after a history of negative experiences with VGP, might allow us to better clarify the role of attitudes in relation to the development of gaming problems. However, as shown in cluster 3 for attitudes, negative attitudes towards gaming do not seem sufficient for CaA to stop playing VGs or at least to stop having problems with them. 

With regard to cognitions towards video games and gaming behaviour, the grouping and interpretation is more complex given that self-dialogue is included in four situations related to VGP. Even so, it was confirmed that the three resulting clusters were significantly related to the level of problematic gaming. The first cluster, “no internal dialogue” (Cognitions-1), presented a very low presence of self-dialogue in the different VGP situations. The second cluster, “moderate internal dialogue” (Cognitions-2), and the third cluster, “high internal dialogue” (Cognitions-3), showed similar response trends with differences in the intensity with which they reacted to VGP situations. It seems that the clusters are ordered according to the intensity of the response to the different situations, or the intensity of the self-dialogue, although there are also some qualitative differences in the responses to the different cases (for example, in the item “it’s great to spend time playing this game”, the scores of the Cognitions-2 and Cognitions-3 clusters were equal). 

In this case, there is also a significant increase in the presence of VGP problems (IGD-9) as the number of the cluster increases (Cognitions-1, -2, -3), with the highest scores in Cognitions-3, as expected [12]. The differences between these three clusters, especially between Cognitions-2 and Cognitions-3, are focused in the intensity of the response, or the action level of the responses, to the VGP situations considered, supporting the idea that people with an addiction do not have an internal dialogue in this regard that is qualitatively different to that of people who do not have a problem [12], in contrast to those who think there is a different type of internal dialogue specifically related to gaming problems [10,31]. The hypothesis of cognitive activation being responsible for certain problems has an empirical tradition in other areas, such as depression or generalised anxiety. This indicates that there are no qualitative differences in rumination or concern compared to that of the general population, but it is instead a matter of the intensity and frequency of thoughts/preoccupations [44,45]. The fact that there are hardly any differences between Cognitions-1 and Cognitions-2 may indicate that an intensity threshold is necessary in the activation levels of responses or self-dialogue for it to translate into differences in gaming problems.

The item “*I am not going to stop until I will achieve it. Better now than tomorrow*”, with higher values in Cognitions-3, clearly broke the parallel behaviour of scores between Cognitions-3 and Cognitions-2. It would be interesting to look further at whether the self-competitive component plays an important role in the establishment of problematic VGP patterns, that is, whether gamers with a moderate cognitive activation and who do not appear to be self-competitive present fewer VGP problems.

When studying possible interactions between the clusters identified and the level of problematic gaming, low values of the different interactions between the clusters of each of the three variables studied were seen. The people who showed greater cognitive activation whilst gaming (Cognitions-3) and also a higher problem awareness (Engagement-3) presented the highest scores in problematic gaming.

The most important result is the high explanatory capacity of these groups of clusters (50% of variance in the scores in gaming problems). It would be interesting to identify whether these figures relate to their importance in the development of VGP problems, or only to their continuance, and also their importance in relation to preventive and/or therapeutic tasks. Of all of them, the engagement clusters have undoubtedly been shown to be the most relevant in this task, followed by the concurrent cognitions with video games.

There are various therapeutic and preventive alternatives for addressing IGD [46,47,48,49]. These results underscore the importance of incorporating psychoeducational modules in therapeutic and preventive plans, focusing on gaming involvement, self-awareness of problems, and cognitive management. This emphasis is not so much on content but on the constant activation of different thoughts related to gaming.

Limitations: This study used a non-hierarchical cluster analysis methodology in which the number of clusters was predetermined. The choice of three clusters was made in order to detect high–average–low or low-intensity–intermediate–intense patterns. Using a different number of clusters could lead to different results that may also be interesting. 

The sample size is not large enough and is limited to a specific geographical area of Spain (Madrid). This could hinder the accurate generalization of results.

Based on previous studies, this work used age and sex as co-variables, but other co-variables such as game type could certainly be included in the model. For example, by using the DSM criteria as a reference for defining IGDs (which are exclusive to online gaming), information about offline gaming is not taken into account. It is deemed appropriate to conduct future studies that also encompass this type of gaming.

Although potential interactions between the identified clusters and their possible association with IGD are explored in the general linear model, a joint analysis of variables (involving implications, attitudes, and cognitions) has not been conducted to identify diverse general profiles of players and their potential relationship with IGD. In order to comprehensively explore this relationship, the use of face-to-face interviews is also considered desirable, as it would enable a more reliable diagnosis of ICD.

With regard to VGP problems, the decision was made to consider these as a continuum instead of categories defined by precise cut-off points. This is because: (1) it allows us to better observe the phenomena by increasing the variability; (2) it does not exclude risk profiles that might not be detected in a category-based approach; and (3) there are no clear cut-off points for the instrument, especially in the Spanish version. However, it is considered important to make progress in this area and for future works to examine the relationships between these categories and the presence of IGD.

## 5. Conclusions

Engagement levels with video games are clearly related to problematic behaviour, and although there is a perception of the problem by the person and the people close to them, this does not seem to stop the development of VGP problems. Also, the presence of problematic gaming behaviours was associated with a high cognitive activation (increased intensity and frequency of self-dialogue). The interactions among engagement, attitudes, and cognitions did not appear to have greater explicated variance of problematic gaming behaviours. 

There were no qualitative internal dialogue differences between people showing more problematic gaming and others. It was a matter of intensity and frequency, and there seems to be an activation threshold of behaviours above which problems appear. 

These results support the design of treatments and approaches that, in addition to being based on awareness and warnings about problems derived from video game abuse, are focused on training and self-control problems with a high cognitive component.

## Figures and Tables

**Figure 1 ijerph-20-07194-f001:**
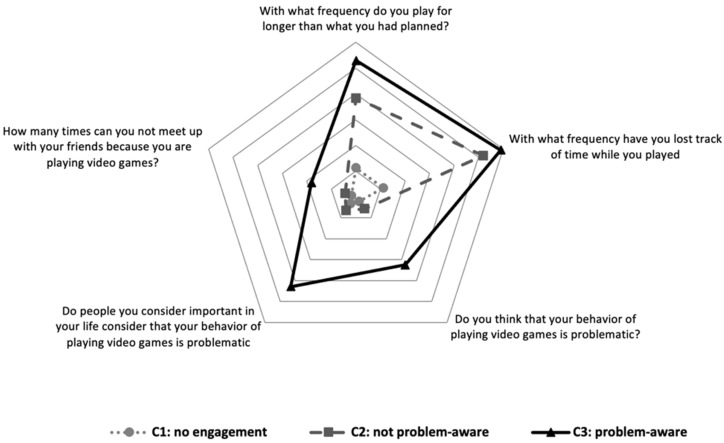
Clusters of engagement with video games.

**Figure 2 ijerph-20-07194-f002:**
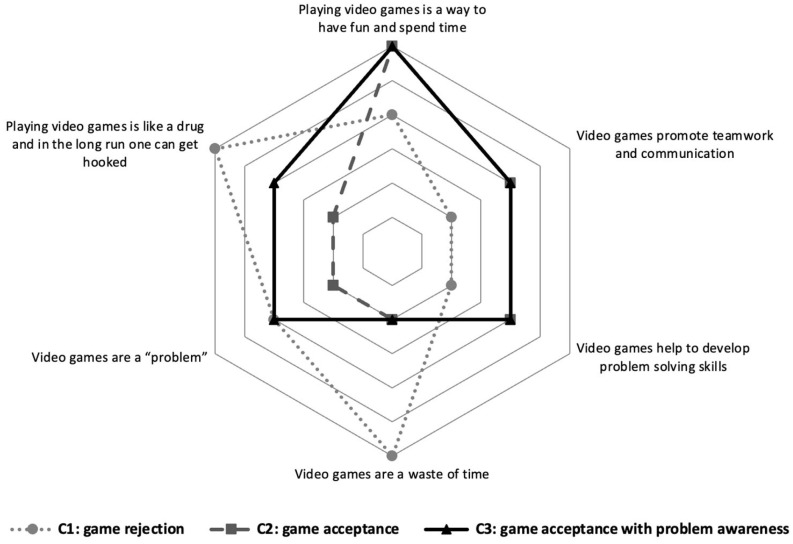
Clusters of attitudes towards video games.

**Figure 3 ijerph-20-07194-f003:**
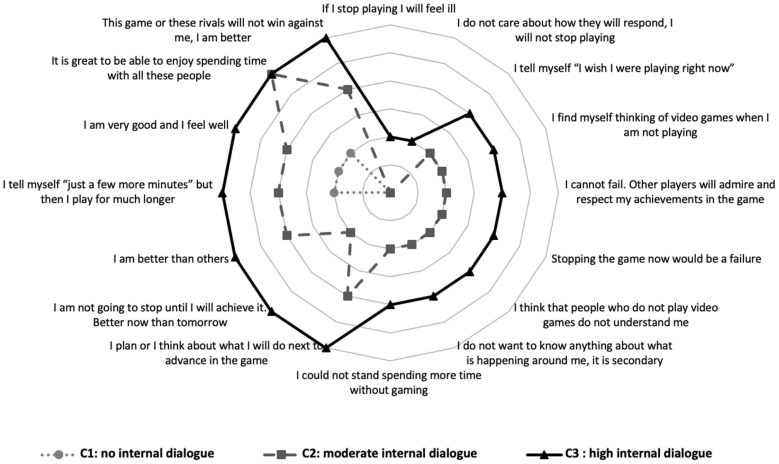
Clusters of cognitions towards video games.

**Table 1 ijerph-20-07194-t001:** Socio-demographic and video gaming usage characteristics of the sample.

*N* = 2887	M (*Sd*)/*n* (%)
Gender	
Male	1659 (57.5%)
Female	1228 (42.5%)
Age	15.35 (2.69)
% use video games	2173 (75.3%)
Number of hours spent each week on video games	6.18 (1.67)
Number of days spent each week on video games	3.41 (2.016)
Preferred device *	
Video consoles	712 (32.8%)
Computer	455 (20.9%)
Mobile phone	655 (30.1%)
Tablet	155 (7.1%)
Others	196 (9%)
Type of video game *	
Action/adventure games	538 (24.7%)
Massively multiplayer online	376 (17.3%)
Sports	323 (14.8%)
Shooters	320 (14.7%)
Others	616 (25.5%)
IGD-9 score	14.58 (5.71)

* Percentage calculated on the 2173 subjects who use video games.; M = mean.

**Table 2 ijerph-20-07194-t002:** Association of each cluster with problematic gambling behaviour.

	IGD-9 Estimated Marginal Means (*Sd*)	*F* (*p*)	Partial *eta*^2^
Engagement E1: no engagement E2: no problem-aware E3: problem-aware	15.34 (0.41) 16.09 (0.20) 21.54 (0.33)	108.28 (*p* < 0.001)	0.091
Attitudes A1: game rejection A2: game acceptance A3: game acceptance with problem awareness	17.56 (0.35) 17.43 (0.23) 17.98 (0.22)	2.48 (*p* = 0.083)	0.002
Cognitions C1: no internal dialogue C2: moderate internal dialogue C3: high internal dialogue	14.49 (0.30) 16.60 (0.18) 21.89 (0.47)	83.96 (*p* < 0.001)	0.072
Engagement × attitudes	--	2.53 (*p* = 0.038)	0.005
Engagement × cognitions	--	7.62 (*p* < 0.001)	0.014
Attitudes × cognitions	--	1.28 (*p* = 0.272)	0.002

Adjusted *R*^2^ = 0.501; estimated marginal means are presented. The model includes age and gender as covariates.

## Data Availability

All databases and files will be available at https://osf.io/nrv45/ (accessed on 1 September 2023).

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
