# Peer review of "Cognitive Patterns and Problematic Use of Video Games in Adolescents: A Cluster Analysis"

_ijerph, 2023, doi:10.3390/ijerph20247194_

Round 1
Reviewer 1 Report
Comments and Suggestions for Authors
The authors performed a cluster analysis to explore the existence of cognitive patterns associated with engagement, attitudes, and concurrent cognitions in adolescents. They also explored the relationship between these patterns and problematic video game playing among 2,173 adolescent game players.
This paper will be worth being published in Int. J. Environ. Res. Public Health. However, several points should be addressed for the further review to reach final acceptance.
The term Pathological Gaming (PG) in the Introduction requires references and brief description about its definition.
The title indicates that the subjects of this study were adolescents and the mean age of the subjects were 15.35 years old. The age of the subjects of this study ranged from 12 to 20 years old. It is uncommon to include 20-year-old youth in adolescents. The authors should define the adolescents clearly.
The subjects were selected from 37 schools in Madrid randomly. The age range of the subjects (12 to 20 y) suggests that these subjects could be heterogeneous including junior high school, high school, and college students. Brief explanation about educational system in Spain will be helpful.
The diagnostic criteria of Internet Gaming Disorder of DSM-5 can include gaming on the internet exclusively while Gaming Disorder of ICD-11 include both online and offline gaming. If the authors have any data about the Internet use of these subjects, they should be referred in this text.
Internet Gaming Disorder Scale–Short-Form (IGDS9-SF) will have cut-off proposed by the developers; 5 or more items answers as ‘5: Very Often’. The authors should report the rate of subjects screened positive according to the cut-off commonly used in other countries despite the lack of reliable data in Spain.
The authors should admit the limitations that the sample size is not large enough and none of the subjects underwent face-to-face interview for the clinical diagnosis of IGD.
Comments on the Quality of English LanguageMinor editing of English language required.
Reviewer 2 Report
Comments and Suggestions for Authors
Title: Fine, clear and informative
Abstract: Has all the elements except limitations [need to add one sentence]
Introduction: Comprehensive review of relevant material - including research into full cluster analysis identifying different types of gamers
Materials and Methods: Good description of sample, but data/stats should be intext or tables, not both
Measurements and data analysis is fine - but need to do a full sample cluster analysis using all variables - two reasons [1] show interactive effects of the three independent variables and [2] allow the reader to get a sense of who are the different types of gamers and the relative influence of the three independent variables
Results: Very clear and interestingly presented in terms of graphics
Discussion: Need to be revised and relocate the results material back into the Results - a discussion should be limited to a brief paragraph summarising the results and then linking the results to other relevant research etc - remove all statistics from discussion
Limitations: At this time this material are not limitations but are justifications of decisions regarding design. A definite limitation [unless authors do a full cluster analysis] is the lack of "profile" of gamers [including a cluster of problematic users]
Conclusion: is really a summary
The material on "treatments and approaches" should have a separate heading and be expanded and embedded in published research on education/interventions
References: Fine
Round 2
Reviewer 1 Report
Comments and Suggestions for Authors
The authors have revised their manuscript following to comments from the reviewers. This current version could be accepted for publication.